# Exclusive breast feeding practice and associated factors among Rural and Urban mothers of child aged 0–6 months in Tahtay Maichew District of Tigray, Ethiopia, 2023/2024: Comparative cross sectional study

**Mebrahtom Birhane Tikue**[1]*, **Teklay Zeru Weldearegawi**[2]*,
**Teklemariam Gebregziabher Goitom**[2], **Berhe Gebrehiwot Tewele**[3],
**Nebiat Desale Gidey**[3], **Willi Bahre**[1]

**1** Department of Nursing, College of Medicine and Health Sciences, Adigrat University, Adigrat, Tigray, Ethiopia, **2** Department of Nursing, College of Health Sciences, School of Nursing, Aksum University, Axum, Tigray, Ethiopia, **3** Araya Kahsu College of Health Science, Axum, Tigray, Ethiopia

* meb2040b@gmail.com, meb5240@gmail.com (MBT); teklayz93@gmail.com (TZW)

## Abstract

### Background

The most well- recognized method of feeding a baby is breastfeeding. Out of all the preventive measures, breastfeeding and supplemental feeding have the most impact on child mortality for infants and young children. Babies are shielded against acute infections by the mother's antibodies found in breast milk, and it boosts a baby's immune system, their reaction to immunizations and cognitive benefits. This was aimed to compare and identify the factors associated with exclusive breastfeeding practice among the rural and urban lactating mothers of the District. Exclusive breast feeding practice of this rural area was in line with the urban which was 68.1% (95% CI, 62.1–73) and 64% (95% CI, 53.5–74.4) respectively. This might be due to the similar primary health care policy intervention system both in the rural and urban areas of the Region. Besides of this, both the rural and urban mothers of the district were more benefited from the Ethiopia primary health care packages.

### Methods

A comparative cross-sectional study was conducted. A total of 346 lactating mothers were enrolled for the study. 5 rural and 1 urban Kebeles were randomly selected. From these the simple random sampling technique was used. The interviewer administered structured questionnaire was employed. Epi-Data Manager Version 4.6 was used to enter, encode, and clean the data for consistency and completeness. The data was analyzed using SPSS version of 22. Bivariable and multivariable logistic regression analysis were performed.

**Data availability statement:** All relevant data are within the manuscript and its Supporting Information files.

**Funding:** The author(s) received no specific funding for this work.

**Competing interests:** The authors have declared that no competing interests exist.

**Acronyms and Abbreviations:** ANC: Ante Natal Care; AOR: Adjusted Odds Ratio; ARI: Acute Respiratory Infection; BF: Breast Feeding; CI: Confidence Interval; COR: Crude Odds Ratio; CS: Caesarean Section; EBF: Exclusive Breast Feeding; EBFP: Exclusive Breast Feeding Practice; EDHS: Ethiopia Demographic Health Survey; EHSTP: Ethiopia Health Sector Transformation Plan; PNC: Post Natal Care; SPSS: Statistical Package For Social Science; TBA: Traditional Birth Attendant; UNICEF: United Nations International Children's Emergency Fund; WHO: World Health Organization.

## Results

A total of 346 lactating mothers were selected in both rural (258) and urban (88) areas of the district. The response rate was 248 (96.12%) in rural and 86 (97.72%) in urban. The prevalence of exclusive breastfeeding practice in the rural and urban areas was 68.1% (95% CI, 62.1–73) and 64% (95% CI, 53.5–74.4), respectively. According to the multivariable logistic regression analysis, in the rural mothers' educational status [AOR = 2.46, 95% CI (1.3–4.6)], parity [AOR = 2.56, 95% CI (1.4–4.7)], and antenatal care visit [AOR = 2.35, 95% CI (1.0–5.4)] were associated factors. In the urban area, education on exclusive breastfeeding practice during antenatal care visits [AOR = 3.46, CI (1.2–10)], mode of delivery [AOR = 3.7, CI (1.2–11.5)] and education on exclusive breast feeding practice during postnatal care visit [AOR = 2.9, CI (1.0–8.1)] were associated factors.

## Conclusion

Based on the mean score, exclusive breastfeeding practice in the rural and urban areas was slightly above the mean. This result doesn't show a significant difference both in the rural and urban areas. But this was lower than the notional and global recommendation level. It was substantially correlated with maternal educational status, parity, and prenatal care visits in the rural areas and with health education during prenatal care visits, method of birth, and postnatal care visits in the urban areas.

## Recommendation

We recommend that health care policy makers and health care providers strengthen the delivery of health education about exclusive breastfeeding practice during ANC visits and PNC visits, and they should advocate institutional delivery. Secondly, mothers should attend ANC and PNC and conduct institutional delivery so as to get information about exclusive breastfeeding practice.

## Background

Breastfeeding is the most well-recognized method of feeding a baby [1]. Out of all the preventive measures, breastfeeding and supplemental feeding have the most impact on child mortality (19%) for infants and early children [1].

If exclusive breastfeeding was made available to all infants under the age of six months, it might avoid 42% of diarrhea, 21% of fever, and 27% of the burden of ARI in children [2]. We can lessen the burden of the three children's morbidities mentioned above by 26% if we guarantee exclusive breastfeeding for the first six months. Once more, wasting was 2.32 times more common in children whose EBF was interrupted between 0 and 3 months of age than in those whose EBF was continued until 6 months of age [2].

Every year, 1.4 million deaths of children under five in the developing countries may be avoided by children under the age of two [3,4]. Non-exclusive breastfeeding raises the risk of acute respiratory infections and diarrhea while also decreasing the amount of nutrients fully absorbed from breast milk [3].

Babies are shielded against acute respiratory infections and diarrhea by the mother's antibodies found in breast milk. Additionally, it boosts a baby's immune system and their reaction to immunizations and has cognitive benefits.

A 90% coverage rate of exclusive breastfeeding would avert around 13% of all deaths among children under five years old [1].

Once more, breastfeeding lowers the risk of type 2 diabetes and ovarian and breast cancers, and it cuts down on postpartum bleeding and lactational amenorrhea (which may be a natural birth control method), and promotes adequate weight recovery for the mother. Benefits to the economy, environment, and mental health are also guaranteed [2,3].

Globally, EBF rates are substantially below what was required to best protect women's and their children's health. Just 41% of babies younger than six months were fed only breast milk. This falls well short of the 70% global objective set for 2030 [1–3]. An estimated 1.4 million fatalities in children under five years old worldwide are attributed to inadequate breastfeeding, with only about 35% of infants dying as a result [1].

Exclusive breastfeeding practice in Sub-Saharan Africa up to six months was estimated to be between 49% and 59% [1]. Exclusive breastfeeding was one of the most prevalent unintentional feeding practices that can lead to newborn death, illness, and malnourishment. This leads to poor weight gain, diarrhea, insufficient breast milk, lactation failure, and an increased vulnerability to infections [3].

A study carried out in Bahr-Dar City reveals that during the first six months of life, children who were not exclusively breastfed had a 2.3-fold higher risk of developing diarrhea compared to those who were [4].

Exclusive breastfeeding practice in the rural and urban areas was affected by different maternal and child socio-demographic, obstetric, and maternal health care service-related factors [5].

A comparative study conducted in Niger shows the coverage of exclusive breastfeeding practice in the urban area (56.8%) was higher than in the rural area (45.7%). This was due to maternal access to education, media, and better behavioral change towards health-seeking behavior [6].

To the reverse, a study in Nigeria showed that rural respondents practiced EBF by 79.8% of the rural respondents compared to the urban (29.0%) of the urban respondents. It was reasoned out that urban mothers were busier at work than the rural mothers [7].

Again, a study conducted in Ethiopia indicated that the practice of exclusive breastfeeding coverage of urban mothers (66.1%) was better than rural mothers' practice (53.5%) [8]. Still, they were showing lower prevalence than that of the WHO recommendation level.

Ethiopia's Ministry of Health has made an effort to improve the best practices for lactating mothers by developing training manuals and training provisions. Despite their incorporation into the primary health care system in accordance with the health extension program, breastfeeding practices are still far from meeting the global recommendation at the 90% level [1].

Research conducted at the international, continental, national, and regional levels indicates that the practice of exclusive breastfeeding was influenced by factors related to mother and child socio-demographics, economics, obstetrics, maternal health care practices, and maternal residential area.

The findings of this research's result will guide the development of intervention techniques for exclusive breastfeeding practice among mothers. Its significance will play a crucial role in the society, helping others who are interested to work in the field of child and maternal health as well as health care providers to assess the effectiveness of their health care delivery system and researchers to conduct additional studies using alternative study designs.

## Method and materials

### Study setting and period

The study was conducted in the Northern Ethiopia, Central Zone of Tahtay Maichew District, which is 1041 kilometers away from Ethiopia's capital, Addis Ababa. The district had a total population of 90,447 (49% male and 51% female), 46,128 females with 21,255 reproductive-age females per year (District report of 2023), and 3111 expected pregnant mothers per year. There were 12 Kebeles. The district was inhabited by Christians and Muslims who practice their respective religions. Public and private health institutions were present in the district, including one public primary hospital, three health centers, twelve public health posts, one private clinic, and three pharmacies. The study was conducted in the Tahtatay Maichew District of Tigray in January 2024.

### Study design

Community based comparative cross-sectional study design was carried out.

### Population

The source population for the study was all lactating mothers with children aged 0–6 months of the district. While the lactating mothers with children aged 0–6 months in the randomly selected Kebeles of the district during the data collection period were the study population.

All lactating mothers with children 0–6 months old during the data collection time were included in the study, whereas mothers who were seriously ill and mothers with seriously ill children were not.

### Sample size determination and sampling procedure

The double population proportion formula in Welaita Zone was used to get the sample size [8]. Using the presumptions listed below, comparative studies were done with a sample size of 55.5% from rural and 66.1% from urban areas, respectively. The investigation yielded the highest sample size in the area, with a margin of error of 5% and a confidence interval of 95%. Using the method for double population proportion, the actual sample size for this study was calculated based on this premise.

$$n = \frac{(Z\alpha + Z\beta)^2 \, (p1q1 + p2q2)}{(P1 - p2)}$$

$$n = \frac{(Z\alpha + Z\beta)^2 \, (p1(1 - p1)) + p2(1 - p2))}{(P1 - p2)}$$

$$n = \frac{(1.96 + 0.84)^2 \, (0.661(1 - 0.661) + 0.555(1 - 0.555))}{(0.661 - 0.55)^2}$$

$$n = 329 + 5\% \text{ nonresponse rate}$$

$$n = 346$$

Where:

Zα is the percentage point of the normal distribution corresponding to the 5% significance level = 1.96.
Zβ is the one-sided percentage point of the normal distribution corresponding to the power of the study at 80% = 0.84.
P 1 = expected proportion (0.661) of the total lactating mothers who practiced exclusive breastfeeding in the urban area and

P2 = expected proportion (0.555) of the total lactating mothers who practiced exclusive breastfeeding in the rural area.

q1 = probability of failure around p1 and q2 = probability of failure around p2.

The total sample of n = **346** was the study subjects by adding a 5% nonresponse rate.

The estimated number of lactating mothers of the district for 2023 was 3111 from the District annual plan. By proportionating with their total reproductive age, the expected lactating mothers of the selected Kebeles was 1553 (Table 1).

These factors for sample size determination were adapted from different literature sources [6,9,10]

## Variables of the study

**Dependent variable.** Mothers' exclusive breast feeding practice status (good or poor).

**Independent variables.** **Variables related to maternal and child socio-demographic characteristics**: age, gender, marital status, place of residence, employment, educational attainment of the mother, ethnicity, religion, spouse's educational level, information access, and the child's age and sex.

**Variables pertaining to health services**: parity, frequency of prenatal visits, attendance at prenatal care services, healthcare professionals' advice on breastfeeding during ANC, and postpartum exclusive breastfeeding instruction.

**Variables related to obstetrics:** birth interval, parity, mode of delivery, place of delivery, and birth attendant.

**Variables related to exclusive breastfeeding practice:** initiation of baby breastfeeding within an hour after birth, offering breast milk only within the first 3 days after birth, provision of colostrum for the child, frequency of breastfeeding >=8 times per 24 hours, and provision of breast milk only up to the date of interviewing.

## Sampling procedure

There were twelve kebeles in the district. Eleven of them were rural, and one was urban. Out of the rural kebelles, five were selected randomly. Therefore, the study participants were taken from May-siye (N = 66; n = 58), Akabi-seat (N = 66; n = 61), May-birazyo (N = 61; n = 54), Debre-shewit (N = 60; n = 53), and Tisha (N = 36; n = 32) (rural Kebelles), and Wukro-maray (N = 99; n = 88) (urban). Where:

N = Proportionate probability of lactating mothers

n = Total Proportionate probability of sample for each Kebeles

Using the lottery method, five rural Kebelles and one town were selected. The total monthly expected number of lactating mothers of the randomly selected Kebeles was 130 mothers/month. The total number of lactating mothers for 3 months was 390.

The sample for each kebele was distributed according to their total population of lactating mothers. The total sample of all the selected Kebeles was 346 respondents. A systematic random sampling procedure was applied for each selected Kebel. K = N/n = 390/346 = 2 was the sampling interval. Then from 1–2 numbers randomly, we picked number 2. Then every kth interval was selected. Therefore, starting from 2, every second volunteer lactating mother from the sample frame

**Table 1. Determinant factors of exclusive breast feeding practice.**

| Factors | % Outcome in un exposed | AOR | Ratio of unexposed to exposed | 5%of non-response rate | N | Reference |
|---|---|---|---|---|---|---|
| Place of delivery | 70.79 | 8.8 | 1 | 5 | 91 | [3] |
| Maternal occupation | 49.55 | 2.57 | 1 | 9 | 179 | [3] |
| Initiation of BF after birth | 30.8 | 4.04 | 1 | 4 | 84 | [2] |
| Mode of delivery | 18.947 | 2.40 | 1 | 12 | 250 | [4] |
| Post natal counseling on EBF | 47.8 | 2.12 | 1 | 13 | 265 | [4] |

**NB.** Confidence level and power were 95% and 80% respectively.

AOR: adjusted odds ratio, N: number of participants and EBF: exclusive breast feeding.

was selected. After that, every volunteer lactating mother in the sample was interviewed in each kebele until the desired sample size was attained.

## Quality control measures

**Data quality control issue.** The quality of the questionnaire about contents, consistency, language, and organization was checked and modified in line with standards, guidance, and comments from advisors and colleagues. A 5% pretest of the questionnaire was administered in Laelay Maichew District in December 2023 before the actual data collection was conducted from December 2023 to February 2024.

**Data collection tool and technique.** The interview-administered structured questionnaire was prepared in the English language and translated to the Tigrigna language, then back to English.

The data collectors were 6 diploma nurses and 2 BSc nurses as supervisors. 2-day training was provided for them. The purpose of the study was explained so as to provide full information for the respondents. The randomly selected mothers were interviewed. Mothers who were in volunteer and absent at the time of data collection were considered non-respondents.

## Data possessing and analysis

Data was entered, encoded, & cleaned for completeness and consistency using Epi-data manager version 4.6 for data entry and was exported to SPSS version 22 for analysis. Bi-variable analysis was used to assess factors' crude OR at $p <= 0.20$ value. Multivariable logistic regression analysis was used for factors with AOR at p values $<= 0.05$ and model fitness was checked with the Hosmer and Leme show test. At a p value that was p (0.708) for the rural and p (0.944) for the urban. Multi collinearity was checked for the independent variables with Spearman correlation; it was $< 0.7$. The P value of 0.05 was considered as statistically significant at the 95% CI (Confidence Interval) level. The results were organized, summarized, and presented using appropriate descriptive measures such as texts and tables.

## Ethical consideration

Ethical clearance was obtained from Aksum University, College of Health Science School of Nursing, and Comprehensive Specialized Hospital, institutional reviewing committee reference number (IRB042/2023), and the District Health Office, and written informed consent was obtained from each study participant, and confidentiality was assured for each participant. In addition, we obtained the consent from the guardians of the minors before conducting the interview

The purpose of that study, the right to refuse participation, and the liberty to refuse or to leave the study at any time were explained to each respondent before the interviewing. To ensure confidentiality, the mother's name was not written on the questionnaire. All methods were performed in accordance with the declarations of Helsinki.

## Result

### Maternal and child Socio-demographic characteristics of the respondents

In all, 346 nursing moms were chosen from the district's rural (258) and urban (88) areas. In both rural and urban areas, the response rate was 248 (96.1%) and 86 (97.72%), respectively. The respondents, who came from both rural and urban areas, ranged in age from 27 to 30 years old (IQR 27–30 years) (Table 2).

### Maternal Health service and obstetric related factors of the respondents

Rural mothers who had antenatal care visits and practiced exclusive breastfeeding were 218 (87.9%). Out of these 156 mothers, 71.6% practice exclusive breastfeeding. Whereas in the urban area, 86 (100%) of them had antenatal care visits. From the antenatal care visited mothers, 54 (65.1%) practiced exclusive breastfeeding (Table 3).

**Table 2. Maternal and child Socio-demographic characteristics of exclusive breast feeding practice and associated factors in Tahtay maichew district 2024 (n = 346).**

| Variable | 1) Rural% (n = 248) | EBFP (%) | 2) Urban% (n = 86) | EBFP (%) |
|---|---|---|---|---|
| Residence | | | | |
| Rural | 248(74.3) | 169(68.1) | | |
| Urban | | | 86(25.7) | 55(64) |
| Maternal age in year | | | | |
| 15-19 | 14(5.6) | 10(71.4) | 4(4.7) | 2(50) |
| 20-24 | 68(27.4) | 44(64.7) | 23(26.7) | 16(69.6) |
| 25-29 | 81(32.7) | 57(70.4) | 34(39.5) | 22(64.7) |
| 30-34 | 57(23) | 41(71.9) | 12(14) | 8(66.7) |
| 35-39 | 23(9.3) | 14(60.9) | 10(11.6) | 5(50) |
| >=40 | 5(2) | 3(60) | 3(3.5 | 2(67) |
| Religion | | | | |
| Orthodox | 248(100) | 169(68.1) | 46(53.5) | 30(69.56) |
| Muslim | | | 18(20.9) | 11(61.1) |
| Catholic | | | 11(12.8) | 8(72.7) |
| Protestant | | | 11(12.8) | 6(54.5) |
| Ethnicity | | | | |
| Tigray | 248(100) | 169(68.1) | 86(100) | 55(64) |
| Marital status | | | | |
| Married | 229(92.3) | 159(69.4) | 78(90.7) | 51(65.4) |
| Divorced | 18(7.3) | 10(55.6) | 8(9.3) | 6(75) |
| Widowed | 1(0.4) | | | |
| Paternal education | | | | |
| Can read and write | 171(74.7) | 123(71.9) | 75(96.2) | 50(66.7) |
| Can't read write | 58(25.3) | 36(62.1) | 3(3.8) | 1(33.3) |
| Paternal occupation | | | | |
| House father | 5(2.2) | 4(80) | 17(21.8) | 11(64.7)) |
| Gov'tal worker | 7(3.1) | 6(85.7) | 18(23.1) | 11(61.1) |
| Trader | 4(1.7) | 4(100) | 10(12.8) | 5(50) |
| Labor worker | 54(23.6) | 35(64.8) | 21(26.9) | 16(76.2) |
| Farmer | 159(69.4) | 110(69.2) | 12(15.4) | 8(66.7) |
| Maternal education | | | | |
| Can read and write | 185(74.6) | 136(73.5) | 72(83.7) | 45(83.7) |
| Can't read and write | 63(25.4) | 33(52.4) | 14(16.3) | 10(71.4) |
| Maternal occupation | | | | |
| House wife | 24(9.7) | 15(62.5) | 29(33.7) | 20(69) |
| Gov'tal worker | 4(1.6) | 3(75) | 16(18.6) | 11(68.8) |
| Trader | 5(2) | 3(60) | 16(18.6) | 10(62.5) |
| Labor | 37(14.9) | 23(62) | 15(17.4) | 9(60) |
| Farmer | 177(71.4) | 124(70.1) | 10(11.6) | 5(50) |
| Other | 1(0.4) | 1(100) | | |
| Source of EBF education materials | | | | |
| Radio | 10(4) | 6(60) | 45(52.3) | 10(18.18) |
| Television | 19(7.7) | 10(52.6) | 16(18.6) | 12(21.81) |

*(Continued)*

**Table 2.** (Continued)

| Variable | 1) Rural% (n = 248) | EBFP (%) | 2) Urban% (n = 86) | EBFP (%) |
|---|---|---|---|---|
| Books | 105(42.3) | 43(40.9) | 9(10.5) | 22(40) |
| None of them | 114(46) | 49(42.98) | 16(18.6) | 11(20) |
| Number of children | | | | |
| 1 | 68(27.4) | 45(66.2) | 20(23.3) | 13(65) |
| 2-3 | 107(43.1) | 80(74.8) | 39(45.3) | 23(59) |
| >=4 | 73(29.4) | 44(60.3) | 27(31.4) | 19(70.4) |
| Child's age in month | | | | |
| Birth −1 | 86(25.7) | 44(71) | 24(27.9) | 18(75) |
| 2-3 | 115(34.4) | 59(67) | 26(30.2) | 16(61.5 |
| 4-6 | 133(39.8) | 65(67) | 36(41.9 | 23(63.9) |
| Child's gender | | | | |
| Male | 118(47.6) | 74(62.7) | 41(47.7) | 26(63.4) |
| Female | 130(52.4) | 95(73.1) | 45(52.3) | 29(64.4) |
| Parity | | | | |
| Primi-para | 88(35.5) | 50(56.8) | 20(23.3) | 11(55) |
| Multipara | 160(64.5) | 119(74.4) | 66(76.7) | 44(66.7) |
| Birth interval of this child with his/her immediate elder | | | | |
| <36month | 73(52.8) | 32(43.83) | 36(54.5) | 23(63.9) |
| >=36month | 87(47.2) | 47(54) | 30(45.5) | 21(70) |

Gov'tal: governmental, EBFP: exclusive breast feeding practice.

## Exclusive breast feeding practice measuring factors

Regarding the mothers' practices in EBF, the mean score of the rural and urban mothers was 68.1% and 64% respectively (Table 4).

## Factors associated with exclusive breast feeding practice

According to the bi-variable analysis, maternal educational status, parity, history of antenatal care visits, place of delivery, and exclusive breastfeeding practice education during postnatal care visits were in the rural mothers. But exclusive breastfeeding practice education during antenatal care visits, mode of delivery, and EBF practice education during postnatal care visits in the urban area were variables having a significance level <=0.20 with exclusive breastfeeding practice, as shown in Table 5.4. According to the multivariable logistic regression analysis of the rural area, maternal educational status, parity, and antenatal care visits were significantly associated factors with exclusive breastfeeding practice. In the urban area, exclusive breastfeeding education during antenatal and postnatal care and mode of delivery were significantly associated factors with exclusive breastfeeding practice.

In the rural area of this study mothers who can read and write were 2.46 times more likely to practice exclusive breast feeding than those who can't read and write[AOR = 2.46, 95% CI (1.3–4.6)]. Regarding to the parity level, multiparous mothers were 2.56 times more likely to practice exclusive breast feeding than those primi-parous [AOR = 2.56, 95%CI(1.4–4.7)].Mothers who got ante natal care visit were practiced exclusive breast feeding 2.35 time more likely than those who didn't get (AOR = 2.35,CI 95% CI (1.0–5.4).

Regarding to the urban area, mothers who got exclusive breast feeding education during their antenatal care visit practiced exclusive breast feeding3.46 times more likely than those who did not get[AOR = 3.46,CI(1.2–10)].Those

**Table 3. Maternal Health service and obstetric related factors of the respondents in Tahtay Maichew district 2024, (n = 346).**

| | 1) Rural (n = 248) % | EBFP (%) | 2)Urban (n = 86)% | EBFP (%) |
|---|---|---|---|---|
| ANC visit | | | | |
| Yes | 218(87.9) | 156(71.6) | 86(100) | 54(65.1) |
| No | 30(12.1) | 13(43.3) | | 1(100) |
| No. ANC visit | | | | |
| >=4times | 13(6) | 11(84.6) | 32(37.2) | 16(50) |
| <4times | 205(94) | 145(70.7) | 54(62.8) | 38(70.37) |
| EBF education during ANC visit | | | | |
| Yes | 119(54.8) | 89(74.8) | 38(44.18) | 30(78.9) |
| No | 99(45.4) | 67(67.6) | 48(55.81) | 24(50) |
| Type of education given during ANC | | | | |
| Start BF within an 1hr after birth | 10(8.40) | 8(80) | 4(10.52) | 2(50) |
| Don't give prelacteal food | 12(10.08) | 7(58.33) | 6(15.78) | 2(33.33) |
| Give your colostrum | 20(16.80) | 17(85) | 7(18.4) | 4(57.14) |
| Give EBF up to 1st six months | 35(29.41) | 26(74.28) | 9(23.68) | 4(44.4) |
| Give breastfeeding>=8x/24hrs | 42(35.29) | 31(73.80) | 12(31.57) | 5(41.66) |
| Place of delivery for this child | | | | |
| Home | 114(46) | 68(59.6) | 6(7) | 4(66.7) |
| Hospital | 134(54) | 101(75.4) | 80(93) | 51(63.8) |
| Mode of delivery if hospital | | | | |
| Spontaneous vaginal | 123(91.8) | 95(77.2) | 61(76.3) | 43(70.5) |
| Caesarean section | 11(8.2) | 6(54.5) | 19(23.8) | 8(80) |
| Delivery assisted if home delivery | | | | |
| TBA | 3(2.6) | 1(33.3) | | |
| Relatives | 111(97.4) | 67(60.4) | 5(83.3) | 4(80) |
| Health professional | | | 1(16.7) | |
| Others | | | | |
| EBF education during PNC | | | | |
| Yes | 82(33.1) | 65(79.3) | 54(62.8) | 40(72.7) |
| No | 166(66.9) | 104(62.7) | 32(37.2) | 14(45.2) |

ANC: antenatal care, NO.ANC: number of antenatal care, TBA: traditional birth attendant, EBF: exclusive breastfeeding and PNC: postnatal care.

who gave spontaneous vaginal delivery practiced exclusive breast feeding 3.7 times more likely than those who didn't get[AOR = 3.7,CI(1.2–11.5)]. Again urban mothers who got exclusive breast feeding education during their postnatal visit practiced exclusive breast feeding 2.9 times more likely than those who didn't get[AOR = 2.9,CI(1.0–8.1)] (Table 5).

## Discussions

In this study we tried to assess exclusive breast feeding practice and associated factors children birth up to six months old in both rural and urban areas which was around two-third. This was in line with the Ethiopia health sector developmental plan V (72%) [9]. But this was lower than the Global recommendations level (90%) [1].

Exclusive breast feeding practice in this rural area was in line with the urban which was 68.1% (95% CI, 62.1–73) and 64% (95% CI, 53.5–74.4) respectively. This might be due to the similar primary health care policy intervention system both in the rural and urban areas of the Region. Besides of this, both the rural and urban mothers of the district were more benefited from the Ethiopia primary health care packages.

**Table 4. Exclusive breast feeding practice measuring factors in Tahtay Maichew district 2024, (n = 346).**

| Variables | | | | |
|---|---|---|---|---|
| | Rural(n = 248) | EBFP (%) | Urban(n = 86) | EBFP (%) |
| Did you put your baby to breast feed within an hour after birth? | | | | |
| Yes | 158(63.7) | | 57(66.3) | |
| No | 90(36.3) | | 29(33.7) | |
| Did you give breast milk only for the child within the first 3 days after birth? | | | | |
| Yes | 226(91.1) | | 81(94.2) | |
| No | 22(8.9) | | 5(5.8) | |
| Did you give your colostrum for this child? | | | | |
| Yes | 238(96) | | 86(100) | |
| No | 10(4) | | | |
| Did you give >=8 times breast feeding within 24 hours? | | | | |
| Yes | 183(73.8) | | 63(73.3) | |
| No | 65(26.2) | | 23(26.7) | |
| Did you give the child breast milk only up to this date? | | | | |
| Yes | 158(63.7) | | 57(66.3) | |
| No | 90(36.6) | | 29(33.7) | |
| Over all EBF practice status | | | | |
| Good | | 169[(68.1), (95%CI,62.1–73)] | | 55[(64) (95% CI, 53.5–74.4)] |
| Poor | | 79(31.9) | | 31(36) |

EBF: exclusive breast feeding, EBFP: exclusive breast feeding practice and CI: confidence interval.

**Table 5. Bi-variable and multivariable logistic regression analyzed factors.**

| 1)Rural (n = 248) | | | 2) Urban (86) | | |
|---|---|---|---|---|---|
| Variable | COR (95%CI) | AOR (95%CI) | Variable | COR (95% CI) | AOR (95% CI) |
| **Maternal education** | | | **EBF education ANC visit** | | |
| Can read & write | 2.52(1.39-4.56) | 2.46(1.3-4.6)** | Yes | 3.28(1.24-8.7) | 3.46(1.2-10)* |
| Can't read and write(1) | | | No(1) | | |
| **Parity** | | | **Mode of delivery** | | |
| Primi-parous (1) | | | Vaginal delivery | 3.285(1.3-9.5) | 3.7(1.2-11.5)* |
| Multiparous | 2.21(1.27-3.83) | 2.56(1.4-4.7)** | Cesarean section(1) | | |
| **Did you get ANC visit** | | | **EBFP education during PNC** | | |
| Yes | 3.29(1.50-7.17) | 2.35(1.0-5.4)* | Yes | 3.24(1.3-8.2) | 2.9(1.0-8.1)* |
| No(1) | | | No(1) | | |
| **Place of delivery** | | | | | |
| Home(1) | | | | | |
| Hospital | 2.1(1.203-3.562) | 1.68(0.82-3.44) | | | |
| **Did you get EBF education during PNC** | | | | | |
| Yes | 0. 44(0.24-0.82) | 1.3(0.6-2.8) | | | |
| No(1) | | | | | |

*P<0.05, **p<0.01 and ***p<0.001

COR: crudes odds ratio, AOR: adjusted odds ratio, ANC: antenatal care, EBF: exclusive breast feeding, EBFP: exclusive breastfeeding practice and PNC: postnatal care

The rural result of this study was higher than the studies conducted in rural areas of Bangladesh, Niger, Ghana, the Welaita Zone, and the Gamo-Gofa Zone of Ethiopia [1,6,8,11,12]. This study was in line with a study conducted in Enderta Woreda [13]. But this was lower than the study conducted in Nigeria [7]. This might be due to the reason that around 36% of rural mothers started additional food before the child was six months old.

Regarding to the urban prevalence of exclusive breastfeeding practice, this was higher than the studies conducted in Nepal, Nigeria, and Addis Ababa [3,7,14]. But it was in line with the study conducted in India, Boditi town of Oromia, and Bahridar [4,8,15]. On the contrary, this result was less prevalent than the study conducted in Dredawa [16]. This might be due to the war's influence. It affected the earlier organized child health care delivery system of the District.

Regarding to the maternal and child socio-demographic characteristics; the result of maternal educational status was consistent with the studies conducted in Amhara Gubalafto District and mini EDHS2019 [2,17]. This might be due to the universal access for education, which was one of the sustainable developmental goals of Ethiopia, which enhanced the level of community awareness on child nutrition. But this was not associated in the urban area. This might be due to urban mothers being highly exposed to mass media influences that enabled them to have better awareness about health-related information. But the problem could be practicing. This could be urban mothers initiating early complementary feeding in order to engage in office or daily labor work.

Concerning with the parity level, it was inconsistent with studies conducted in the Wolega Zone, the Gamo-Gofa Zone, and rural Ethiopia [12,18,19]. This might be due to multiparous mothers having more exposure during the child nutrition programs provided by the different stakeholders of child health. Again, this was inconsistent with the urban area of this study. The reason might be rural mothers lack access to more communication channels about child care, and they have less experience sharing about exclusive breastfeeding practices.

Regarding to obstetrics and maternal health service-related factors, a history of antenatal care visits was consistent in mothers who had it in the study conducted in rural Ethiopia [19]. This might be due to the similarity in socio-demographic and economic status and health policy. But this was inconsistent with a study conducted in the Gamo-Gofa Zone [12]. This might be due to the variation in commitment of the health care actor's commitment and motivation and some socio-cultural differences. This was inconsistent with the urban area. It might be due to rural mothers being far from the health facilities, lack of audio-visual channels that propagate information about maternal and child health, and lack of optimal health service in their nearby health facilities due to the post-war crisis.

Exclusive breastfeeding education during antenatal care follow-up in the urban area was consistent with the study conducted in Addis Ababa [3].This might be due to the proxy measure of receiving greater exposure to information from different media channels or better utilization of health services with staff better trained on infant feeding. This was inconsistent with the rural area. It might be due to rural mothers being eager to apply the instructions provided by their health care provider since they lack communication channels about exclusive breastfeeding.

In this study spontaneous vaginal delivery in the urban area was consistent with the study conducted in Niger and Addis Ababa [3,6]. This could be due to the pain the mother experiences and may delay the giving of breast milk and start other formulas or cow milk in the first few days to the baby and greater exposure to information about exclusive breastfeeding from health professionals. This was inconsistent with the rural area. The reason could be most urban mothers give institutional delivery and then prefer cesarean delivery to shorten labor pain.

Regarding to the EBF education during postnatal care visits; mothers who got education were more practiced than those who didn't get it. This result was consistent with a study result conducted in Boditi Town of the Welaita Zone [13]. This could be due to counseling mothers during PNC visits, which may enhance mothers' understanding and appreciation of the demands and benefits of EBFP. Besides this, they may have the opportunity to get further information about the benefits of EBFP for the baby from the health care providers and their peers. The result of this was inconsistent with the rural. The reason could be urban mothers interfere with exclusive breastfeeding practice to engage in work. Then they practice additional feeding mechanisms like provision of cow or formula milk or early ignition of complementary feeding.

## Conclusion

In the rural and urban areas, exclusive breastfeeding was practiced by around two-thirds. But this was still lower than the global (90%) recommendation level. Maternal educational status, parity and ANC visits in the rural area, and EBF practice education during antenatal care visits, mode of delivery, and PNC visits in the urban area were significantly associated with exclusive breastfeeding practice. This result shows no significant difference in both areas. But, this signals the health care providers to modify their plan so as to achieve the global level.

## Strength and Limitation

This study was comparing mothers' practice of exclusive breastfeeding in the rural and urban areas. But this study didn't assess the impact of economic status; the respondents' bias considering the study had an effect with food aid.

## Recommendations

We recommend that health care policy makers and health care providers strengthen the delivery of health education about exclusive breastfeeding practice during ANC visits and PNC visits, and they should advocate institutional delivery. Secondly, mothers should attend ANC and PNC and conduct institutional delivery so as to get information about exclusive breastfeeding practice. Lastly, researchers should conduct cohort and case-control study designs to identify the impact of exclusive breastfeeding practice on mothers and children.

## Author contributions

**Conceptualization:** Mebrahtom Birhane Tikue, Teklemariam Gebregziabher Goitom.

**Data curation:** Mebrahtom Birhane Tikue, Teklemariam Gebregziabher Goitom, Nebiat Desale Gidey.

**Formal analysis:** Mebrahtom Birhane Tikue.

**Investigation:** Mebrahtom Birhane Tikue.

**Methodology:** Mebrahtom Birhane Tikue, Teklay Zeru Weldearegawi, Teklemariam Gebregziabher Goitom, Berhe Gebrehiwot Tewele, Nebiat Desale Gidey.

**Project administration:** Mebrahtom Birhane Tikue.

**Software:** Mebrahtom Birhane Tikue.

**Supervision:** Teklay Zeru Weldearegawi, Teklemariam Gebregziabher Goitom, Berhe Gebrehiwot Tewele, Willi Bahre.

**Validation:** Mebrahtom Birhane Tikue.

**Visualization:** Teklemariam Gebregziabher Goitom, Willi Bahre.

**Writing – original draft:** Mebrahtom Birhane Tikue.

**Writing – review & editing:** Mebrahtom Birhane Tikue, Teklay Zeru Weldearegawi, Teklemariam Gebregziabher Goitom, Berhe Gebrehiwot Tewele, Nebiat Desale Gidey, Willi Bahre.

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
