## [Decision Letter · Decision Letter 0]

20 Sep 2025

PONE-D-25-42029
Exclusive breastfeeding practice and associated factors among Rural and Urban mothers of child aged 0-6 months in Tahtay Maichew District of Tigray, Ethiopia, 2023/2024: Comparative cross-sectional study
PLOS ONE

Dear Dr. Tikue,

Thank you for submitting your manuscript to PLOS ONE. After careful consideration, we feel that it has merit but does not fully meet PLOS ONE’s publication criteria as it currently stands. Therefore, we invite you to submit a revised version of the manuscript that addresses the points raised during the review process.

We look forward to receiving your revised manuscript.

Kind regards,

Kahsu Gebrekidan, Ph.D.

Academic Editor

PLOS ONE

Journal Requirements:

Reviewers' comments:

Reviewer's Responses to Questions

**Comments to the Author**

1. Is the manuscript technically sound, and do the data support the conclusions?

Reviewer #1: Yes

Reviewer #2: Yes

2. Has the statistical analysis been performed appropriately and rigorously?

Reviewer #1: Yes

Reviewer #2: Yes

3. Have the authors made all data underlying the findings in their manuscript fully available?

Reviewer #1: Yes

Reviewer #2: Yes

4. Is the manuscript presented in an intelligible fashion and written in standard English?

Reviewer #1: Yes

Reviewer #2: No

5. Review Comments to the Author

Reviewer #1: Hello dear authors.

MS Id: PONE-D-25-42029

Title: Exclusive breastfeeding practice and associated factors among Rural and Urban mothers of child aged 0-6 months in Tahtay Maichew District of Tigray, Ethiopia, 2023/2024: Comparative cross-sectional study

Type: Research Article

Here are my recommendations about the mentioned MS:

Title:

• Looks good.

Abstract:

• Looks good.

Introduction:

• Strengthen your introduction with more references.

Methodology:

• Setting, period, population, and sampling are to be in one section, and remove any extra information that is not directly related to the method of the study.

• The method includes two mentions of sampling. You could be described in one paragraph.

• The tool's reliability has to exist.

• Dedicate a paragraph and describe variables for the tools.

• The pilot study and the reliability process for the tools have not been mentioned.

• Please consider removing the operational definition of the terms from the method section, as it may not be necessary.

Results:

• Comments for table two must just exist and contain the majority.

Discussion:

• No need to rewrite the results in the discussion.

Conclusion:

• Looks good.

References:

• Looks good.

Figures and tables:

• No figure exists.

Reviewer #2: Dear Authors, I want to extend my appreciation for your research, which provides valuable insights and addresses the critical issue of exclusive breastfeeding practices in a comparative context. Below are some comments and suggestions:

1. Abstract

*Consider emphasizing the significance of the comparative aspect

*It’s good to see the use of logistic regression for analysis but including a brief explanation of the significance of these analyses would be beneficial for readers unfamiliar with statistical methods.

*Line 42 and 43: “VII In the urban area education on exclusive breast feeding practice during antenatal care visit pra” revise this as it’s not clearly written.

2. Background

*The background provides comprehensive information about the importance of exclusive breastfeeding. Consider organizing it into clearly defined flow.

3. Method and materials

*The sample size determination process is described, but it would be beneficial to provide the specific formula used for clarity. Additionally, ensure that the rationale behind the assumed proportions is clearly explained.

*Line 207 and 208: “K= N/n =390/346=1.12. Using lottery method we picked randomly number 3. Every Kth interval was selected” How did you used the Kth interval while it is mentioned that K=1.12?

*The sampling procedure is detailed, but it might be helpful to summarize the key steps in a flowchart for better clarity and readability.

4. Result

*The socio-demographic characteristics are well-presented. Consider highlighting any significant differences between rural and urban respondents that could impact the study's conclusions.

5. Discussion

*The discussion presents important findings, but the flow can be improved by using clearer transitions between key points. Consider grouping related themes together for better coherence.

*How could educational interventions be tailored to meet the needs of urban versus rural mothers?

6. Conclusion

*Consider including specific recommendations for health interventions or future research based on the study's findings.

6. PLOS authors have the option to publish the peer review history of their article (what does this mean?). If published, this will include your full peer review and any attached files.

Reviewer #1: No

Reviewer #2: No

---

## [Author Response · Author response to Decision Letter 1]

29 Nov 2025

Response letter for reviewers

Oct 03, 2025

Dear Editor,

Thank you for giving us the opportunity to submit the revised draft of our manuscript entitled “Exclusive breast feeding practice and associated factors among Rural and Urban mothers of child aged 0-6 months in Tahtay Maichew District of Tigray, Ethiopia, 2023/2024: Comparative cross sectional study” to PLOS ONE. We appreciate the time and effort that you and the reviewers have dedicated for providing your valuable feedback on our manuscript. We are grateful to you and the reviewers for the insightful comments on our paper. We have tried to revise our manuscript in accordance with the suggestions and comments provided by you and the reviewers.

Here is a point-by-point response to the comments made by the reviewers, the editor, and the editorial staff.

Sincerely,

Mebrahtom Birhane Tikue

Adigrat University, Ethiopia

E-mail: meb2040b@gmail.com/meb5242@gmail.com

Phone number: +251901297441/251974570700

Part One: Point-by-point responses to editor and editorial staff

A rebuttal letter that responds to each point raised by the academic editor and reviewer(s). You should upload this letter as a separate file labeled 'Response to Reviewers.

Response: Uploaded

A marked-up copy of your manuscript that highlights changes made to the original version. You should upload this as a separate file labeled 'Revised Manuscript with Track Changes.

Response: A revised manuscript with track changes is submitted in accordance with the instruction.

An unmarked version of your revised paper without tracked changes. You should upload this as a separate file labeled 'Manuscript.

Response: A revised manuscript without tracked changes is submitted in accordance with the instruction.

Response: It meets PLOS ONE’s style requirements.

2. We suggest you thoroughly copyedit your manuscript for language usage, spelling, and grammar.

Response: Modified

Response: Incorporated

4. Please include captions for your Supporting Information files at the end of your manuscript, and update any in-text citations to match accordingly.

Response: Modified

Response: It meets PLOS ONE’s style requirements.

Response: Yes, I obtained the consent from the participants; the format consent was incorporated in the questionnaire prior to starting the interviewing.

Please confirm at this time whether or not your submission contains all raw data required to replicate the results of your study. Authors must share the “minimal data set” for their submission. PLOS defines the minimal data set to consist of the data required to replicate all study findings reported in the article, as well as related metadata and methods.

If your submission does not contain these data, please either upload them as Supporting Information files or deposit them to a stable, public repository and provide us with the relevant URLs, DOIs, or accession numbers. Response: incorporated

5. PLOS requires an ORCID ID for the corresponding author in Editorial Manager on papers submitted after December 6th, 2016. Please ensure that you have an ORCID ID and that it is validated in Editorial Manager. To do this, go to ‘Update my Information’ (in the upper left-hand corner of the main menu), and click on the Fetch/Validate link next to the ORCID field. This will take you to the ORCID site and allow you to create a new ID or authenticate a pre-existing ID in Editorial Manager. Response: Yes, obtained

Part two: Point-by-point responses to reviewers

Reviewer 1

Thank you, dear reviewer, for reviewing our paper. We have answered each of your points below.

1) Strengthen your introduction with more references.

Response: Dear reviewer, thank you for your valuable feedback. We have already incorporated.

2) Setting, period, population, and sampling are to be in one section, and remove any extra information that is not directly related to the method of the study. Response: Dear reviewer, thank you for your valuable feedback .We have already modified it.

3) The method includes two mentions of sampling. You could be described in one paragraph. Response: Dear reviewer, thank you for your constructive and valuable feedback. We have already modified it as a paragraph.

4) The tool's reliability has to exist.

Response: Dear Reviewer, thank you very much for your constructive and valuable feedback. We would like to clarify that we conducted a pre-test on 5% of the sample and made the necessary modifications based on the results. However, we did not report Cronbach’s alpha because it is primarily recommended for Likert-scale items, and in tools with dichotomous or non-Likert items, the values often appear low despite the tool functioning well.

5) Dedicate a paragraph and describe variables for the tools.

Response: Dear reviewer, thank you for your constructive e feedback. We modified it.

6) The pilot study and the reliability process for the tools have not been mentioned.

Response: Dear reviewer, thank you for your constructive and valuable feedback. We would like to clarify that we conducted a pre-test on 5% of the sample and made the necessary modifications based on the results.

7) Please consider removing the operational definition of the terms from the method section, as it may not be necessary.

Response: Dear reviewer, thank you for your constructive and valuable feedback. We removed them from page 11 and 12.

8) Comments for table two must just exist and contain the majority.

Response: Dear reviewer, thank you for your constructive and valuable feedback. We have already modified it.

9) No need to rewrite the results in the discussion.

Response: Dear reviewer, thank you for your constructive and valuable feedback. We have already modified it.

10) No figure exists.

Response: Dear reviewer, thank you for your constructive and valuable feedback. But we did not plan to display results with figures. Ruther we preferred to display them with tables and paragraphs.

Part two: Point-by-point responses to reviewers

Reviewer 2

Thank you, dear reviewer, for reviewing our paper. We have answered each of your points below.

1. Abstract

1.1) Consider emphasizing the significance of the comparative aspect

Response: Dear reviewer, thank you for your constructive and valuable feedback. We modified it by adding additional explanations.

1.2) It’s good to see the use of logistic regression for analysis but including a brief explanation of the significance of these analyses would be beneficial for readers unfamiliar with statistical methods.

Response: Dear reviewer, thank you for your constructive and valuable feedback. We modified it.

1.3) Line 42 and 43: “VII In the urban area education on exclusive breast-feeding practice during antenatal care visit pra” revise this as it’s not clearly written.

Response: Dear reviewer, thank you for your constructive and valuable feedback. They are already modfied.

2. Background

2.1) The background provides comprehensive information about the importance of exclusive breastfeeding. Consider organizing it into clearly defined flow.

Response: Dear reviewer, thank you for your constructive and valuable feedback. We incorporated comparative explanation about this study.

3. Method and materials

3.1) The sample size determination process is described, but it would be beneficial to provide the specific formula used for clarity. Additionally, ensure that the rationale behind the assumed proportions is clearly explained.

Response: Dear reviewer, thank you for your constructive and valuable feedback. We have incorporated both using the double population proportion and factor based sample size determination. Then we chose the formula that gave us the largest sample size which was the double population proportion sample size determination method.

3.2) Line 207 and 208: “K= N/n =390/346=1.12. Using lottery method we picked randomly number 3. Every Kth interval was selected” How did you used the Kth interval while it is mentioned that K=1.12?

Response: Dear reviewer, thank you for your constructive and valuable feedback. We have already modified 1.2 to 2 and then then we picked number 2 randomly from 1and 2. Then very Kth volunteer lactating mother from the sample from were selected for interviewing.

3.3) The sampling procedure is detailed, but it might be helpful to summarize the key steps in a flowchart for better clarity and readability.

Response: Dear reviewer, thank you for your constructive and valuable feedback. We incorporated the sampling procedure in the form of flow chart/figure form in the indicated place.

4. Result

4.1)The socio-demographic characteristics are well-presented. Consider highlighting any significant differences between rural and urban respondents that could impact the study's conclusions.

Response: Dear reviewer, thank you for your constructive and valuable feedback. We have already incorporated it.

5. Discussion

5.1) The discussion presents important findings, but the flow can be improved by using clearer transitions between key points. Consider grouping related themes together for better coherence.

Response: Dear reviewer, thank you for your constructive and valuable feedback. We modified it by paraphrasing the concepts.

5.2) How could educational interventions be tailored to meet the needs of urban versus rural mothers?

Response: Dear reviewer, thank you for your constructive and valuable feedback. Creating maternal behavioral change and awareness can increase the health seeking behavior. As a result, it increases mothers to attend and searching health related information from different Medias. Thus, help them differentiate the advantages and disadvantages of exclusive breast feeding.

6.Conclusion

6.1) Consider including specific recommendations for health interventions or future research based on the study's findings.

Response: Dear reviewer, thank you for your constructive and valuable feedback. We incorporated list of recommendations for the concerned bodies.

---

## [Decision Letter · Decision Letter 1]

14 Dec 2025

PONE-D-25-42029R1
Exclusive breastfeeding practice and associated factors among Rural and Urban mothers of child aged 0-6 months in Tahtay Maichew District of Tigray, Ethiopia, 2023/2024: Comparative cross-sectional study
PLOS One

Dear Mr. Mebrahtom,

Thank you for submitting your manuscript to PLOS ONE. After careful consideration, we feel that it has merit but does not fully meet PLOS ONE’s publication criteria as it currently stands. Therefore, we invite you to submit a revised version of the manuscript that addresses the points raised during the review process.

We look forward to receiving your revised manuscript.

Kind regards,

Kahsu Gebrekidan, Ph.D.

Academic Editor

PLOS One

Journal Requirements:

Reviewers' comments:

Reviewer's Responses to Questions

**Comments to the Author**

1. If the authors have adequately addressed your comments raised in a previous round of review and you feel that this manuscript is now acceptable for publication, you may indicate that here to bypass the “Comments to the Author” section, enter your conflict of interest statement in the “Confidential to Editor” section, and submit your "Accept" recommendation.

Reviewer #1: (No Response)

2. Is the manuscript technically sound, and do the data support the conclusions?

Reviewer #1: Partly

3. Has the statistical analysis been performed appropriately and rigorously?

Reviewer #1: Yes

4. Have the authors made all data underlying the findings in their manuscript fully available?

Reviewer #1: Yes

5. Is the manuscript presented in an intelligible fashion and written in standard English?

Reviewer #1: Yes

6. Review Comments to the Author

Reviewer #1: Results:

• Comments for table three and table four are inadequate. Please report in the table comments all significant and majority results.

• Define abreviations and symbols used in the tables below the table (for all table).

Discussion:

• No need to rewrite the results in the discussion if need just report p values or percentages.

Conclusion:

• Mention the brief summary of your findings and main implications of your results.

7. PLOS authors have the option to publish the peer review history of their article (what does this mean?). If published, this will include your full peer review and any attached files.

Reviewer #1: No

---

## [Author Response · Author response to Decision Letter 2]

4 Feb 2026

Point by point feedback to Reviewer #1:

Results:

1) Comments for table three and table four are inadequate. Please report in the table comments all significant and majority results.

Response: Dear reviewer, thank you for your constructive and valuable feedback. We have already modified it,

2) Define abbreviations and symbols used in the tables below the table (for all table).

Response: Dear reviewer, thank you for your constructive and valuable feedback. We have already modified them.

Discussion:

1) No need to rewrite the results in the discussion if need just report p values or percentages.

Response: Dear reviewer, thank you for your constructive and valuable feedback. We have already modified it.

Conclusion:

1) Mention the brief summary of your findings and main implications of your results.

Response: Dear reviewer, thank you for your constructive and valuable feedback. We have already modified it.

---

## [Editor Report · Decision Letter 2]

18 Feb 2026

Exclusive breastfeeding practice and associated factors among Rural and Urban mothers of child aged 0-6 months in Tahtay Maichew District of Tigray, Ethiopia, 2023/2024: Comparative cross-sectional study

PONE-D-25-42029R2

Dear Dr. Mebrahtom,

We’re pleased to inform you that your manuscript has been judged scientifically suitable for publication and will be formally accepted for publication once it meets all outstanding technical requirements.

Kind regards,

Kahsu Gebrekidan, Ph.D.

Academic Editor

PLOS One
---

## [Editor Report · Acceptance letter]

PONE-D-25-42029R2

PLOS One

Dear Dr. Tikue,

I'm pleased to inform you that your manuscript has been deemed suitable for publication in PLOS One. Congratulations! Your manuscript is now being handed over to our production team.

Kind regards,

on behalf of

Dr. Kahsu Gebrekidan

Academic Editor

PLOS One